# Predictive Power of In Silico Approach to Evaluate Chemicals against *M. tuberculosis*: A Systematic Review

**DOI:** 10.3390/ph12030135

**Published:** 2019-09-16

**Authors:** Giulia Oliveira Timo, Rodrigo Souza Silva Valle dos Reis, Adriana Françozo de Melo, Thales Viana Labourdette Costa, Pérola de Oliveira Magalhães, Mauricio Homem-de-Mello

**Affiliations:** 1InSiliTox, Department of Pharmacy, Faculty of Health Sciences, University of Brasilia, Brasilia 70910-900, Brazil; 2Laboratory of Natural Products, Department of Pharmacy, Faculty of Health Sciences, University of Brasilia, Brasilia 70910-900, Brazil

**Keywords:** *Mycobacterium tuberculosis*, tuberculosis, in silico, virtual screening, docking

## Abstract

*Mycobacterium tuberculosis* (Mtb) is an endemic bacterium worldwide that causes tuberculosis (TB) and involves long-term treatment that is not always effective. In this context, several studies are trying to develop and evaluate new substances active against Mtb. In silico techniques are often used to predict the effects on some known target. We used a systematic approach to find and evaluate manuscripts that applied an in silico technique to find antimycobacterial molecules and tried to prove its predictive potential by testing them in vitro or in vivo. After searching three different databases and applying exclusion criteria, we were able to retrieve 46 documents. We found that they all follow a similar screening procedure, but few studies exploited equal targets, exploring the interaction of multiple ligands to 29 distinct enzymes. The following in vitro/vivo analysis showed that, although the virtual assays were able to decrease the number of molecules tested, saving time and money, virtual screening procedures still need to develop the correlation to more favorable in vitro outcomes. We find that the in silico approach has a good predictive power for in vitro results, but call for more studies to evaluate its clinical predictive possibilities.

## 1. Introduction

According to the latest World Health Organization (WHO) report, tuberculosis (TB) is still one of the top 10 causes of death and the leading cause from a single infectious agent (above HIV/AIDS). In 2017 alone, TB caused approximately 1.3 million deaths among HIV-negative people and additional 300,000 deaths among HIV-positive patients [1]. Added to this framework, multidrug-resistant TB (MDR/TB) and extensively drug-resistant TB (XDR/TB) have been increasing over the years as a result of spontaneous mutations in the genome of *M. tuberculosis* and the emergence of those mutants as the dominant strain, resulting in a loss of effect of first and second lines of anti-TB drugs, like Rifampicin and Isoniazid [2]. There is a global need for the development of a new set of pharmaceutical medicines that can address this alarming issue.

Developing a new drug traditionally requires at least 10 years of extensive research and funding; however, the employment of an initial computer-aided drug design (CADD) or in silico approach can decrease that [3]. In silico drug screening, as a virtual shortcut, is used to assist researchers in choosing between millions of molecules from different databases and studying their affinity to known targets, helping them to exclude molecules that would have no biological action and thus optimize their research [4].

In silico drug screening can be divided into two main paths, and the accuracy of the results obtained depends on the amount of information available about both ligand and targeted structure [5]. Ligand-based drug screening relies on the data available about inhibitors that will be studied in a number of methods, such as quantitative structure-activity relationship (QSAR) [6], pharmacophore models, and 2D (chemical) [7]/3D (shape) similarity search [3]. The structure-based drug screening (SBDS) requires that a 3D shape of the structure is accessible and that you have a feasible docking program. It uses docking tools such as GLIDE [8], AutoDock [9], or GOLD [10], among others, to screen a large database library of compounds, such as ZINC [11], Maybridge [12], and Chembridge [13], to dock the targets on postulated inhibitors and identify hit molecules through docking score analysis.

To further refine the obtained in silico results, it is often necessary for researchers to perform an in vitro or in vivo assay based on their virtual hit compounds, such as enzyme inhibition (IC50) [14], minimum inhibitory concentration (MIC) [15], and cytotoxicity assay [16]. This step is essential to certify that what they found in silico is in sync with the in vitro assays, has minor cytotoxicity, and may be developed into novel drugs.

In silico tools are often used as evidence for some biological effect previously evaluated. Molecular docking, specifically, has several issues such as the lack of complete mechanisms involving the ligand‒protein interactions. Flexible docking is a way to better characterize the interactions, but remains poor at providing full protein flexibility information [17]. Scoring functions predict the binding capabilities, but there is still no consensus that could improve its affinity predictions [18,19,20]. Since this methodology is confirmatory, not essentially predictive, we chose to exclude studies that only performed in silico approaches to endorse biological results previously obtained.

Based on this background, this article aimed to collect all the research published until 15 August 2018 that performed at least one of the in silico methods cited above and corroborated the results with an in vitro or in vivo assay, succeeding at a critical analysis of the obtained results. This systematic review may be useful for new researchers looking for novel ligands or targets inside *M. tuberculosis* and will support research that is already in progress.

## 2. Results and Discussion

### 2.1. Mycobacterium tuberculosis Enzyme Targets

There were 29 different proteins targeted from all manuscripts analyzed. The most exploited Mtb enzyme was Enoyl-[acyl-carrier-protein] reductase (NADH) (EC 1.3.1.9). It was used as a target in nine of the 46 articles analyzed (~19.6%). DNA topoisomerase I (EC 5.6.2.2) and DNA topoisomerase II (EC 5.6.2.3) were evaluated three times (~6.5%) each. Third place was also a tie between Shikimate kinase (EC 2.7.1.71) and DNA ligase (NAD (+)) (EC 6.5.1.2), both used two (~4.4%) times each. All the other enzymes were targeted one time (~2.2% each) and are shown in Table 1. Four manuscripts did not aim for enzymes: two analyzed an iron-dependent regulator (IdeR) gene [21,22] (with PDB coding for 1U8R for both of them), one analyzed a Filamenting temperature-sensitive mutant (FtsZ) protein [23] (with PDB coding for 1RLU), and one made a pharmacophore-based QSAR, comparing isoniazid with derivatives [24].

It is well established that proteins with a resolution over 3 ångström (Å) have a lower quality and show only basic contours of the protein chain, so the interaction between the structures must be inferred, leading to a higher chance of error [25]. Hence, choosing a PDB with more than 3 Å of resolution may decrease the predictive power of the in silico method, because one cannot see the exact electron density map. Knowing that, we found only three manuscripts that utilized proteins with a resolution above 3 Å [6,9,26]. All other documents used proteins with a resolution lower than 3 Å and are described in Table 1.

From the 26 different enzymes evaluated, 11 (42%) were transferases (EC 2.-.-.-), five (19%) were ligases (EC 6.-.-.-), four (15%) were oxidoreductases (EC 1.-.-.-), and the remaining six were isomerases (3; 12%; EC 5.-.-.-), hydrolases (2; 8%; EC 3.-.-.-), and lyases (1; 4%; EC 4.-.-.-). Translocases (EC 7.-.-.-) were not used in any of the selected articles.

Enoyl-[acyl-carrier-protein] reductase (NADH), coded by the *InhA* gene, was the most studied enzyme. This is the key enzyme in the biosynthesis of mycolic acid and composition of the mycobacterial cell wall [27]. It catalyzes the NADH-specific reduction of 2-trans-enoyl-ACP, essential for fatty acid elongation [28] and is the main target of Isoniazid (INH) [29] and Ethionamide (ETH) [30], first and second line treatments, respectively. These are pro-drugs that need activation by the catalase peroxidase (KatG/EthA) encoding enzyme to form the inhibitory INH/ETH‒NAD adduct. However, mutations in KatG/EthA have been reported and correlated with the development of Mtb resistance to INH and ETH [31,32], thus leading to the need for new molecules that can either directly inhibit the InhA enzyme or find novel molecular paths that can ultimately promote bacterium eradication. Within the 46 manuscripts, the Enoyl-[acyl-carrier-protein] reductase (NADH) (EC 1.3.1.9) was evaluated nine times [10,15,33,34,35,36,37,38,39]. All analyzed articles were able to correctly dock their postulated inhibitors on different sites of the enzyme and studied the obtained results with in vitro assays. This demonstrates that there is a global set of efforts in the search for novel InhA inhibitors and promising products, which will be discussed further.

DNA topoisomerases (I and II) are the main enzymes associated with DNA replication, transcription, recombination, and chromatin remodeling. They work by introducing temporary single- or double-strand breaks in the DNA to directly modulate DNA topology [40,41]. The major difference between them is that while type I topoisomerase induces transient single-strand breaks in the DNA, type II topoisomerase induces double-strand cleaving [42]. Their mechanism of action has long been known, and they have been a target for both anticancer drugs, such as camptothecins (as type I is found in all eukaryotes), and antibacterial drugs, such as quinolones [43]. All this background has made the DNA topoisomerases a satisfying target for novel anti-TB drugs. The mycobacterial DNA topoisomerase II, namely DNA gyrase (GyrB), was the second most exploited enzyme, seen four times [44,45,46,47]. It was observed, with the aid of ligand and structure-based screening, that novel inhibitors could be seen to be targeting this enzyme. In third place, the *Mycobacterium tuberculosis* DNA topoisomerase I (Mttopo I) was targeted three times and it was demonstrated that in silico methodologies can successfully be used to screen for novel molecules [48] and repurpose former approved drugs, such as amsacarine [49], imipramine, and norclomipramine [50].

It is notable that, within the 46 papers, we found 29 distinct targets with different effects on bacterium survival. Despite increasing evidence of Mtb resistance, there are many efforts in the search for novel targets. However, few drugs are actually being released into the pharmaceutical market. After decades of stasis in the development of new anti-TB drugs, bedaquiline and delamanid were the ones most recently approved for tuberculosis treatment [51]. The U.S. Food and Drug Agency (FDA) has approved bedaquiline for MDR-TB and delamanid as a compassionate care option for XDR-TB and TDR-TB infections. Also, the European Medicines Agency (EMA) has approved both agents for MDR-TB. The lack of novel drugs, added to evidence of hERG toxicity and the lipophilicity of the only recent medicines available [52], supports the need for studies of both new Mtb targets and ligand inhibitors.

### 2.2. PDB, Organisms, and Expression System

We found 40 different PDBs used in the 46 evaluated manuscripts. The most applied PDB was 4B6C, which was seen four times (8%) [44,45,46,47]. 4B6C codes for the structure of *M. smegmatis* GyrB ATPase domain in complex with an aminopyrazinamide, a known inhibitor of the GyrB [71]. The use of *M. smegmatis* as a surrogate for *M. tuberculosis* has been previously demonstrated in the literature, as a result of Mtb’s low specific activity [72] and slow growth [71]. Also, Saxena et al. [45] did a sequence alignment of the *M. smegmatis* and *M. tuberculosis* DNA GyrB and found that they share 87.4% similarity. This crystallized protein was used as a structural framework for structure-based virtual screening of databases [44,46], molecular docking studies [47], and structure‒activity relationship (SAR) assay [45].

Subsequently, 4U0J was identified three times (6%) [10,34,36], coding for the crystal structure of *Mycobacterium tuberculosis* enoyl reductase (INHA) complexed with 1-cyclohexyl-5-oxo-n-phenylpyrrolidine-3-carboxamide refined with new ligand restraints [73]. It was used to screen virtual libraries in the search for novel inhibitors [10,34] and to study the binding interactions between developed inhibitors [36].

In the sequence, 1U8R codes for an IdeR (iron-dependent regulator) protein complexed with DNA and was evaluated two times (4%) [21,22]. 2IYQ (Shikimate kinase complexed with ADP), 1WE2 (shikimate kinase in complex with MGADP and shikimic acid) and 2IYZ (Shikimate kinase in complex with shikimate-3-phosphate and ADP) were all studied twice by the same group (4% each) [13,54]. Finally, 1ECL (7K N-terminal fragment of *E. coli* DNA topoisomerase I), 1MW8 (complex between *E. coli* DNA Topoisomerase I and Single-Stranded DNA) and 1MW9 (another *E. coli* DNA Topoisomerase I and Single-Stranded DNA) were also studied two times each (4%) [49,50]. All other PDBs were analyzed one time (2% each) and are described in Table 1.

This finding means that the crystal structures of a determined protein are useful for both structure- and ligand-based screening and can be used to analyze the binding affinities of a discovered ligand as a means of studying the interaction between the atoms—thereby developing novel scaffolds for lead optimization, enhancing its properties against a target.

### 2.3. Virtual Screening Methods Applied

To facilitate the analysis, we divided the virtual screening methods into two main pathways. As cited above, there are two central routes that can be followed when screening for novel molecules, a ligand-based drug screening and a structure-based drug screening; we also studied a combination of both. Ligand-based drug screening was applied in 21 of the 46 manuscripts (46%). Structure-based drug screening was used in 19 papers (41%). A combination of both methods was seen in the six (13%) remaining articles. The referenced information is presented in Figure 1.

The ligand-based drug screening method is based on the hypothesis that “similar molecules have similar activity” [74]. In this case, researchers first design a common pharmacophore within the inhibition ability moiety, to search for similar molecules that ultimately may have a similar biological activity. Structure-based drug screening uses the structure of a known target to screen a database for inhibitors.

After the evaluation of all 46 documents, we found that there was a balance between the presence of both methods, as ligand-based drug screening was observed in 21 manuscripts and structure-based drug screening was identified in 19 documents. Although it is said that merging the two methods leads to better outcomes, only six manuscripts applied ligand- and structure-based methods to boost their predictive powers (Figure 1).

### 2.4. Databases Screened

After analyzing all manuscripts, we retrieved 15 different databases used in virtual screening. Of the 46 papers, 14 did not use any database (26%). The most explored screening database was ZINC, found in eight of the manuscripts (15%). In second place, there are six unknown in-house databases, which were not openly cited (11%). Maybridge and Chembridge were both screened four times each (almost 8% each). BITS Pilani and Asinex were seen three times each (accounting 6% each). A collaborative drug discovery database was used two times (4%). All other databases were screened one time (2% each) and are displayed in Figure 2.

It is noteworthy that, although some researchers did not screen any database, more than one database was sometimes screened in the same manuscript, as a means of checking for further molecules.

A database can be used to screen both structures of targeted proteins and small molecule inhibitors. Although this screening process is very common when applying an in silico method, we found that 14 documents did not screen any database. All 14 manuscripts designed and synthesized their proposed ligands and docked them to verify the binding affinities. Also, we observed five in-house databases that were not explicitly cited, which might reduce the predictive abilities by not accounting for the largest number of molecules possible, thereby decreasing the reproducibility. However, the most used database was ZINC, found in eight of the manuscripts. ZINC is a free public database with over 230 million purchasable compounds in ready-to-dock format with the latest updates available at ZINC’s website [75]. It has custom subsets that can be created, edited, shared, docked, and downloaded. Manuscripts that use ZINC as the ligand library usually select molecules inside some category, like “in-stock subset (about 12 million compounds),” to obtain faster and better results, using a fraction of all available compounds.

The second most used database was a tie between Chembridge [6,10,13,34] and Maybridge [12,65,67,70]. The Chembridge database is composed of over 1.3 million target-focused compounds and it was a pioneer in the 3D-pharmacophore-based diversity library back in 1995. Also, it has implemented nuclear magnetic resonance (NMR) quality control of all of its compounds and is said to have a high-end purchasable custom library, available at Chembridge’s website [76]. On the other hand, Maybridge is the specialist in organic compounds, acknowledging over 53,000 compounds that generally follow Lipinski’s rule of five and have a good ADME profile, available to purchase at Maybridge’s website [77].

The aims of this analysis were to explain that choosing a database to screen depends solely on the objective and resources of one’s research and to highlight the pros and cons of the one most frequently used.

### 2.5. Docking Software Employed

We found 13 different types of docking software being used for binding studies. The leading software was Glide, which was used 20 times (35%). Autodock versions 3 and 4 come in second, applied nine times (19%). GOLD software was used seven times (12%). FlexX and Surflex Dock were both used four times (accounting for 7% each). Autodock VINA, DOCK, LibDock, and FRIGATE were used two times each (3% each). All other docking software was used once (2% each) and is shown in Figure 3.

Glide and Autodock have been compared in the past with the least cited ones, such as FlexX and Surflex, and found to be equally good [78]. Also, when Glide was compared to GOLD, the third most popular software, it was found to be overall a superior choice for docking [79] and to have superior performance when identifying binding modes [80]. Admittedly, comparing docking programs is difficult because of the many factors that have to be taken into consideration (such as the similar sizes of binding-site regions and the time required to distinguish them) [81], but the findings of this review seem to be consistent with previous published articles, and we found Glide and Autodock to be good docking tools with high predictive abilities, followed by GOLD, FlexX, and Surflex.

### 2.6. In Vitro or In Vivo Testing

As it was a prerequisite in our initial analysis, only manuscripts that did posterior in vitro or vivo assays were selected. To standardize the outcome, we systematized the methodologies that were commonly employed by the researchers. The three most used in vitro assays were minimum inhibitory concentration (MIC), applied 30 times; enzymatic inhibition (measuring IC_50_), which was used 25 times; and cytotoxicity assay, employed 20 times. All other in vitro assays were specific to the hypothesis the researchers were trying to prove (such as tuberculosis-infected macrophage assay [39] or intracellular killing of *M. tuberculosis* [63]) and were all described at least one time each within 19 manuscripts [9,12,13,15,21,22,33,39,44,48,49,50,53,54,57,58,62,63,67,70].

The MIC analysis is by definition the minimum concentration of a determined compound required to completely inhibit bacterial growth. Therefore, we noted whether the in silico methodologies were accurate for predicting the best possible MIC (which would be the lowest value) and whether the authors employed a control with known inhibitors to support the obtained results. Also, to systematize the outcome, we found the ratio between the best MIC obtained and the controls analyzed, such as isoniazid, rifampicin, ethambutol, etc. For all those manuscripts that did not apply a reference compound [9,22,35,49,50,53,56,57,65,69,70], we calculated the mean MIC value (0.78 µM) of Isoniazid (the most potent *M. tuberculosis* inhibitor), using the values found in our chosen manuscripts, after outlier exclusion (z-score higher than 3). The molecules were considered excellent if they had a MIC ratio value below or close to 1, meaning that the compound found was equally effective to the control, or more so. We found that 11 manuscripts presented excellent MIC, superior to at least one of the tested/calculated controls, and 18 had average MIC, not superior to the tested controls. Information about compound names, MICs, ratio values, IC_50_, and docking scores is given in Table 2.

Only one manuscript we found did an in vivo assay. Sridevi et al. (2015) [48] performed an in vivo anti-mycobacterial experiment using a zebrafish model infected by *Mycobacterium marinum*, and their most potent inhibitor (3b) was found to be more effective than isoniazid and rifampicin, which is in line with their in vitro MIC results. Also, they did an enzymatic inhibition of their synthesized compounds and found that molecule 3b was again the most potent, with IC_50_ of 2.9 µM and docking score of −5.62 (Glide).

The second most applied in vitro assay was enzymatic inhibition by measuring the IC_50_ of a determined molecule. A few manuscripts have done enzymatic inhibition assays, but not calculated the IC_50_, so they were not accounted for in this specific analysis [12,36,38,45]. The inhibitory concentration (IC_50_) determines the concentration of a certain compound needed to inhibit 50% of a targeted enzyme, meaning that the lowest IC_50_ value will refer to the most potent compound. Among all 46 retrieved manuscripts, 25 did enzymatic inhibition studies by calculating the IC_50._ The most promising compounds were found by Jeankumar et al. (2014) and Saxena et al. (2015) [44,47], with docking scores of −5.88 and −10.62 by Glide software, respectively.

Despite the controversial outcome of this review analysis, it was found that the in silico approach was indeed able to yield good in vitro outcomes, but not to the highest degree. Surprisingly, the results might mean that, although prior in silico methods drastically decreased the number of tested molecules, hence saving time and money, some adjustments may be required to further restrain and recover scaffolds that can maintain in vitro efficacy, not just present good docking scores. Additionally, it is necessary to continue encouraging studies, as even those manuscripts with good outcomes did not perform in vivo assays, an essential step in developing a new drug.

Additionally, a cytotoxicity assay was performed. This assay does not concern the predictive power of the in silico methodologies analyzed in this review, but it can be predicted by ADMEt (absorption, distribution, metabolism, excretion, and toxicology) in silico methods, such as the OSIRIS drug discovery informatics system [82]. Nevertheless, it is important that approved drugs have few or nontoxic effects in human cellular lines, so it was an interesting feature to study. We found that only 19 of the analyzed manuscripts did cytotoxicity assays of their most promising molecules [10,12,13,15,16,21,23,33,34,39,44,47,48,54,56,57,63,65,70]. The other 26 documents did not test their best molecules for toxic effects.

### 2.7. Validation Procedures

Validation of the virtual screening procedures or in silico methodologies is not mandatory, but it is often seen. Despite this, of the 46 papers, only 19 did any kind of assay to validate their results [6,11,12,13,14,15,16,23,33,35,38,39,44,46,47,55,56,58,61]. Validation trials were considered if they performed a redocking experiment with the targeted protein and its original ligand, compared the binding conformations of the found molecules and the original ligands on the targeted protein, and performed a molecular dynamics simulation. Molecular dynamics (MD) simulation uses conformational ensembles, which are a better representation of real macromolecules, as they estimate flexibility and dynamic properties [83]. This method can predict and validate the virtual screening processes since it can more efficiently simulate biological mechanisms. Although validation procedures may enhance the predictability of a determined method, there is not a cause‒effect relationship in terms of having a good outcome and having performed any kind of validation, so it might remain optional.

### 2.8. Timeline Analysis of Retrieved Manuscripts

As it was our aim to spotlight all manuscripts published until 15 August 2018 that met our eligibility criteria, we also did a timeline analysis of the retrieved documents. Through our screening process, we retrieved 46 documents, ranging from 2005 to 2018, which are described in Figure 4. Two manuscripts were from 2005 [9,53]; one each was from 2007 [60], 2008 [64], and 2009 [65]; two each were from 2011 [34,55] and 2012 [36,63]; five were from 2013 [10,26,35,38,70]; 10 were from 2014 [14,16,23,44,45,46,49,58,66]; nine were from 2015 [6,11,12,15,47,48,50,56,67]; six were from 2016 [13,33,54,57,61,69]; five were from 2017 [21,22,39,59,68]; and finally, only two were from 2018 [37,62].

Following the analysis, we wanted to see whether there was an increase in the application of in silico methodology to predict in vitro activity over time. In Figure 4 it is clear that there was a rise on the number of manuscripts that followed these steps from 2005 to 2015. This most likely means that the reputation of the in silico method prior to in vitro assays improved over time. Also, we analyzed which ones performed a virtual screening procedure to see whether there was an increase in compliance with this specific method. It was found that the use of this method followed the growth of in silico methods in general. This is evidence that scientists accept in silico methodologies in the early steps of developing a new drug, and as part of the development of new and interesting ligand databases and scoring functions. We also found a slight decrease from 2016 to 2018, which may be explained by the end date of the screening process (15 August) and the constraints placed on the course of the research. The limited dataset presumably caused us to lose a few documents that followed these steps but were published after the chosen date. Also, by administering a very stringent screening process (starting with more than 2000 manuscripts and getting down to only 46), we might have failed to retrieve a couple of documents. Nevertheless, the fences we put on this screening made us able to collect the exact information needed.

## 3. Materials and Methods

### 3.1. Background Definitions

The search method employed in this systematic review aimed to include studies that used docking or any other virtual screening method in the search for new molecules against *Mycobacterium tuberculosis* and tested it in vitro and/or in vivo.

### 3.2. Data Sources and Searches

Three different databases were applied to conduct a comprehensive survey: PubMed (National Library of Medicine—https://www.ncbi.nlm.nih.gov/pubmed), Science Direct (Elsevier—https://www.sciencedirect.com/), and Web of Science (Thomson Reuters Scientific—https://www.webofknowledge.com/).

Search terms were chosen to find any published research article that initially used a virtual screening method to search for new molecules against tuberculosis, and then performed an in vitro or in vivo assay to validate the screening, so that we could later apply further exclusion criteria. Search terms “mycobacterium tuberculosis” [MeSH Terms] OR “mycobacterium” AND “tuberculosis” OR “mycobacterium tuberculosis” AND “docking” were used. Searches were conducted using the final limit date of 15 August 2018. Results from all databases provided 2424 manuscripts, divided according to Figure 5. After duplicate removal, 1645 articles were left.

Results were exported to reference management software (EndNote™, Thomson Reuters, Toronto, ON, Canada), in which all selections and analyses were performed.

### 3.3. Study Selection

Firstly, as it was our intent to retrieve manuscripts that performed assays using only *Mycobacterium tuberculosis*, we searched titles and abstracts for the term “Mycobacterium tuberculosis,” resulting in 1149 manuscripts.

Secondly, using EndNote^TM^, we searched the abstracts for the terms “docking” or “docked” or “QSAR” or “virtual screening,” which decreased the results to 836 manuscripts.

Finally, we searched this list for the terms “in vitro” or “in vivo” or “IC50” or “EC50” or “MIC” in abstracts. That resulted in a semifinal set of 273 documents. In the last round, the 273 manuscripts were divided between all authors for total paper reading and exclusion of those that did not match the inclusion criteria. In this round we selected 72 manuscripts, which were then discussed by the authors. After a critical evaluation, 40 manuscripts were selected for the analysis. During data collection, we were able to retrieve six extra manuscripts from the references cited in the 40 manuscripts that for some reason did not appear on our first screening process (from the databases). After submission, a reference found during the reviewing process was still included, to make a final total of 46 manuscripts.

The workflow was summarized using a PRISMA diagram, recommended for systematic reviews and meta-analysis [84], and is displayed in Figure A1 (Appendix A).

### 3.4. Data Extraction Process

The following information was extracted from all the included studies: (a)*Mycobacterium tuberculosis* enzymes target, EC code, and accepted nomenclature [85](b)PDB, organism and expression system [86](c)Virtual screening methods applied (if applied)(d)Databases screened (if applied)(e)Docking software (if applied)(f)In vitro or in vivo assay(g)Validation procedure (if applied)(h)Years in which manuscripts were published

## 4. Conclusions

Preliminary virtual screening methods are able to aid researchers looking for novel targets and help them to rank the best scoring compounds, drastically decreasing the number of undesirable scaffolds. However, when chosen for in vitro assays, only a few of the novel targets retain biological activities and are suitable for further study, regardless of the virtual methods, databases, and docking software applied. This surprising outcome means that in silico methodologies need to be further explored to yield better outcomes, but their use is recommended, particularly in the early steps of developing new pharmaceutical drugs against *Mycobacterium tuberculosis*.

## Figures and Tables

**Figure 1 pharmaceuticals-12-00135-f001:**
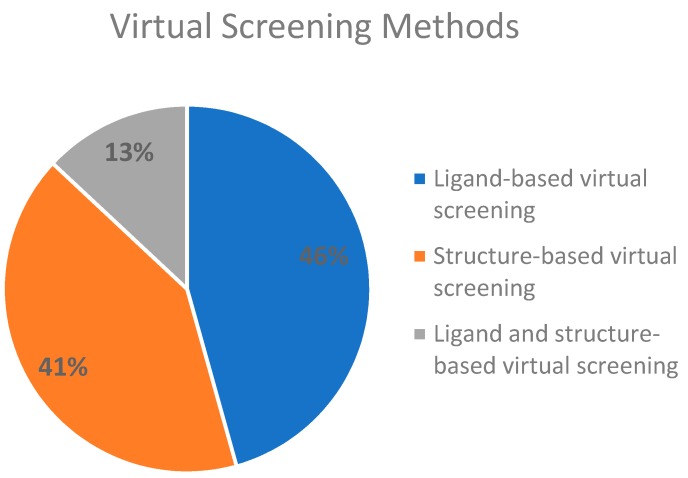
Virtual screening (VS) methods used on the analyzed manuscripts. References for ligand-based VS: [6,11,12,13,22,23,35,36,37,39,45,47,54,55,58,59,60,62,68,69]; structure-based VS: [9,10,14,16,26,33,34,44,46,48,49,50,53,56,57,61,63,64,65] and both ligand and structure-based VS [15,21,38,66,67,70].

**Figure 2 pharmaceuticals-12-00135-f002:**
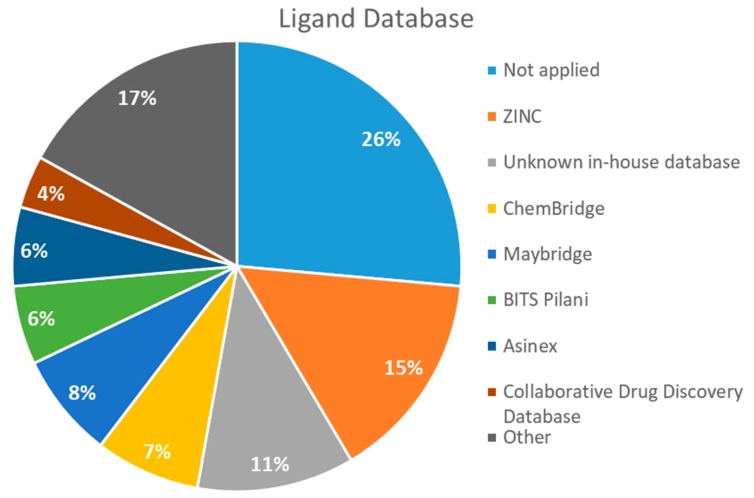
The ligand databases used most for screening. References: No database [15,22,23,33,35,36,37,39,47,54,58,62,68,69]; ZINC [11,21,26,38,55,63,66,67]; in-house [9,45,53,57,64]; Chembridge [6,10,13,34]; Maybridge [12,65,67,70]; BITS Pilani [44,48,56]; Asinex[14,46,48]; Collaborative Drug Discovery Database [49,50]; Other (CDRI [60], FDA approved drugs [67], ChemDiv [6], ChemSpider [59], TAACF [16], GSK [16], Enamine REAL [61], NCI [21] and Specs [70]).

**Figure 3 pharmaceuticals-12-00135-f003:**
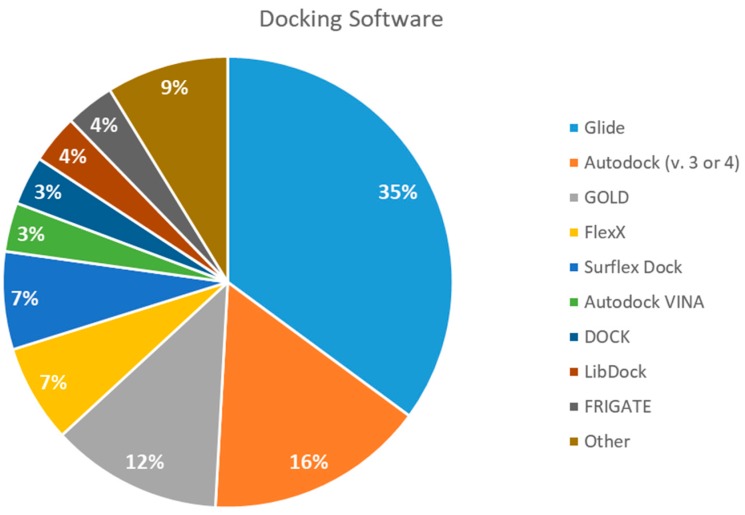
The most used docking software. References: Glide [6,11,13,14,15,16,36,39,44,46,47,48,56,62,68,69,70]; Autodock (v.3 or 4) [9,12,21,22,37,38,53,54,59,64,67]; GOLD [9,10,14,26,34,38,56]; FlexX [12,60,62,65]; Surflex Dock [12,23,33,62]; Autodock VINA [31,47]; DOCK [34,57]; LibDock [49,50]; FRIGATE [55,63] and Other (Biopredicta [58]; Surflex-geomx [12]; CDOCKER [61]; SABRE [66]; Argus Dock [35]).

**Figure 4 pharmaceuticals-12-00135-f004:**
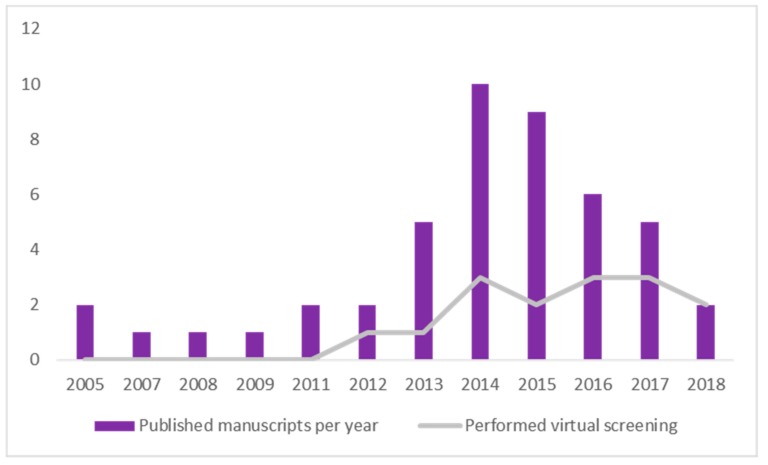
Timeline analysis of published papers. The y axis represents the number of evaluated manuscripts,—46 in total. The x axis represents the years in which the cited documents were published, from 2005 to 2018. The gray line represents the number of manuscripts that performed virtual screening within the published documents in that year.

**Figure 5 pharmaceuticals-12-00135-f005:**
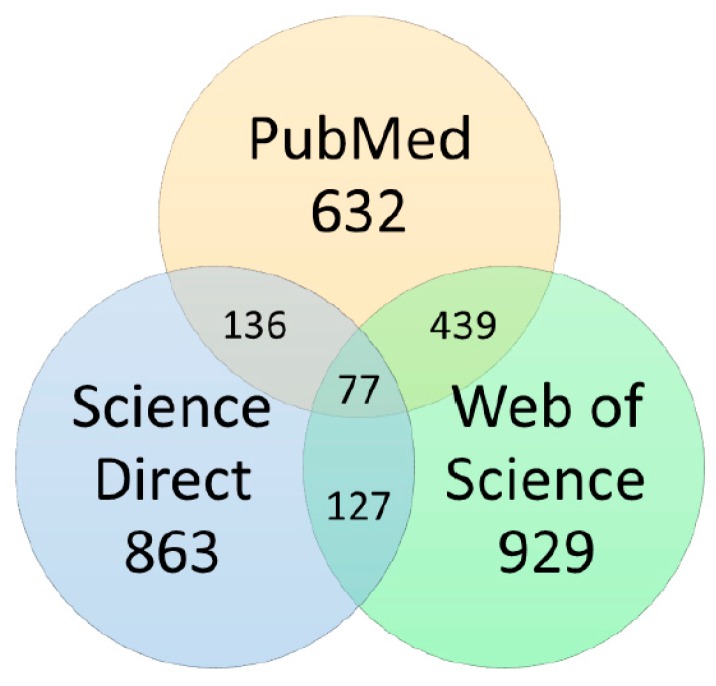
Number of manuscripts found in primary search according to the database searched. Duplicate and triplicate records are shown as the circles’ intersections.

**Table 1 pharmaceuticals-12-00135-t001:** *Mycobacterium tuberculosis* enzymes targeted, PDBs utilized, resolution of 3D targets (Å), references obtained, EC code for enzymes, and number of times the respective enzyme was found.

Enzyme Targeted	PDB	Resolution (Å)	References	EC Code	Quantity
Enoyl-[acyl-carrier-protein] reductase (NADH)	4U0J	1.62 Å	[10,34,36]	EC 1.3.1.9	9
	4TZK	1.62 Å	[35]		
	2NSD	1.90 Å	[39]		
	2AQ8	1.92 Å	[37]		
	3FNG	1.97 Å	[33]		
	1P45	2.60 Å	[15]		
	1P44	2.70 Å	[38]		
DNA topoisomerase (ATP-hydrolyzing)	4B6C	2.20 Å	[44,45,46,47]	EC 5.6.2.3	4
DNA topoisomerase I	1MW9	1.67 Å	[49,50]	EC 5.6.2.2	3
	1ECL	1.90 Å	[49,50]		
	1MW8	1.90 Å	[49,50]		
	3PX7	2.30 Å	[48]		
DNA ligase (NAD (+))	1TAE	2.70 Å	[53]	EC 6.5.1.2	2
	1ZAU	3.15 Å	[9]		
Shikimate kinase	2IYQ	1.80 Å	[13,54]	EC 2.7.1.71	2
	1WE2	2.30 Å	[13,54]		
	2IYZ	2.30 Å	[13,54]		
Diacylglycerol O-acyltransferase	5KWI	1.30 Å	[55]	EC 2.3.1.20	1
3-dehydroquinate dehydratase	2Y71	1.50 Å	[56]	EC 4.2.1.10	1
Leucine-tRNA ligase	2VOC	1.50 Å	[57]	EC 6.1.1.4	1
6,7-dimethyl-8-ribityllumazine synthase	2C92	1.60 Å	[58]	EC 2.5.1.78	1
Dihydropteroate synthase	1EYE	1.70 Å	[59]	EC 2.5.1.15	1
Chorismate mutase	2F6L	1.70 Å	[60]	EC 5.4.99.5	1
D-glutamyltransferase	3TUR	1.72 Å	[61]	EC 2.3.2.-	1
Pantoate-beta-alanine ligase (AMP-forming)	3IVX	1.73 Å	[16]	EC 6.3.2.1	1
3-oxoacyl-[acyl-carrier-protein] reductase	1UZN	1.91 Å	[62]	EC 1.1.1.100	1
dUTP diphosphatase	1MQ7	1.95 Å	[63]	EC 3.6.1.23	1
L-lysine 6-transaminase	2CIN	1.98 Å	[64]	EC 2.6.1.36	1
dTMP kinase	1W2H	2.00 Å	[65]	EC 2.7.4.9	1
Alanine dehydrogenase	2VHW	2.00 Å	[14]	EC 1.4.1.1	1
Thermitase	4HVL	2.00 Å	[66]	EC 3.4.21.66	1
3-deoxy-7-phosphoheptulonate synthase	2B7O	2.30 Å	[11]	EC 2.5.1.54	1
1,4-alpha-glucan branching enzyme	3K1D	2.33 Å	[67]	EC 2.4.1.18	1
NAD (+) synthase (glutamine-hydrolyzing)	3DLA	2.35 Å	[68]	EC 6.3.5.1	1
Pantothenate kinase	3AF3	2.35 Å	[62]	EC 2.7.1.33	1
o-succinylbenzoate-CoA ligase	5C5H	2.40 Å	[69]	EC 6.2.1.26	1
Nonspecific serine/threonine protein kinase	2PZI	2.40 Å	[12]	EC 2.7.11.1	1
Acetolactate synthase	1N0H	2.80 Å	[70]	EC 2.2.1.6	1
Thioredoxin-disulfide reductase	2A87	3.00 Å	[26]	EC 1.8.1.9	1
Proteasome endopeptidase complex	2FHG	3.23 Å	[6]	EC 3.4.25.1	1

**Table 2 pharmaceuticals-12-00135-t002:** Compound nomenclatures, MIC (µM) obtained from manuscript assays, MIC ratio (compound MIC/Control MIC), IC50 (µM) obtained from manuscript assays, docking scores with software applied and the respective references.

Compound	MIC (µM)	MIC Ratio (Cmp/Ctrl)	IC50 (µM)	Docking Score (Software)	Reference
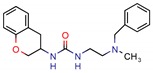 G7650246	-	-	35.3	NS (Glide)	[6]
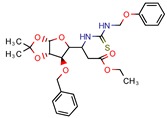 2	14.7	18.8 (Isoniazid ^1^)	4.0	−14.4 kcal/mol (Autodock)	[9]
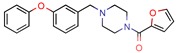 KES4	-	-	4.8	83.5 (GoldScore)	[10]
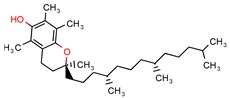 Alpha-tocopherol	-	-	21.0	−7.2 (Glide Score)	[11]
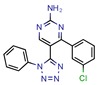 NRB04248	-	-	-	5.7 (Surflex Score)−20.0 KJ/mol (FlexX)−9.5 Kcal/mol (Autodock)	[12] *
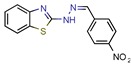 5489375	-	-	10.7	NS (Glide)	[13]
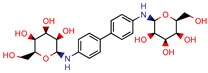 Lead 1	-	-	35.5	−9.9 (Glide Score)	[14]
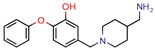 4h	80.0	219. 5 (Isoniazid)	-	−9.1 (Glide Score)	[15]
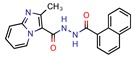 5b	4.53	6.3 (Isoniazid)	1.9	−8.6 (Glide Score)	[16]
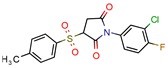 I-108	45.8	754.1 (Rifampicin)	63.6	−6.2 kcal/mol (Autodock)	[21]
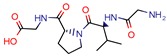 GVPG 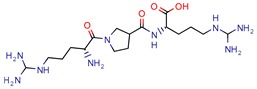 RPR	200.0	256.4 (Isoniazid ^1^)	-	−5.28 Kd (GVPG)−5.78 Kd (RPR) (Autodock)	[22]
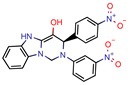 7b	7.3	49.8 (Isoniazid)	-	3.9 (Surflex Score -logKd)	[23]
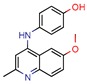 1	-	-	12.5	Consensus using GoldScore, Chemscore and ASPscore (GOLD)	[26]
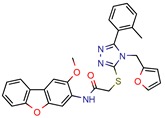 I1	-	-	5.3	83.0 (GoldScore)	[34]
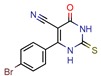 2g	-	-	-	−5 to −6 (Glide-XP Score)	[36] *
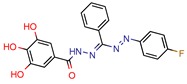 C9	2.0	2.8 (Isoniazid)0.3 (Ethambutol)0.9 (Ofloxacin)	3.4	−9.5 (Glide Score)	[39]
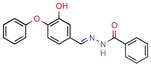 DE3	8.5	0.4 (Isoniazid)	-	7.0 (Surflex Score -logKd)	[33]
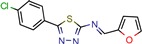 Fb 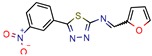 Fe	10.710.3	13.713.2 (Isoniazid ^1^)	-	−9.3 kcal/mol (Fb) −9.2 kcal/mol (Fe) (Argus Dock)	[35]
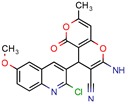 PA	4.0	0.04 (Pyrazinamid)	-	−9.0 kcal/mol (Autodock VINA)	[37]
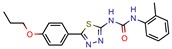 ZINC09137707	-	-	-	NS GoldScore (GOLD)	[38] *
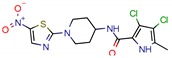 14	7.5	11.4 (Isoniazid)	0.5	−5.9 (Glide Score)	[44]
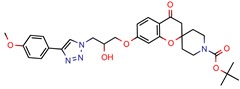 Ex-355	-	-	-	NP	[45] *
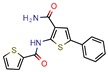 Lead 11	-	-	1.5	62.1 (GoldScore)−10.3 (Glide XP Score)	[46]
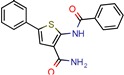 23	4.8	7.3 (Isoniazid)21.0 (Rifampicin)2.2 (Ofloxacin)0.6 (Ethambutol)	0.8	−10.6 kcal/mol (Glide)	[47]
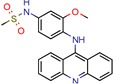 m-AMSA	125.0	160.3 (Isoniazid ^1^)	-	94.6 (Libdock Score 21–150)	[49]
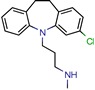 Norclomipramine	60.0	76.9 (Isoniazid ^1^)	-	95.2 (Libdock Score ~46.4–126.3)	[50]
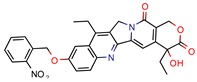 3b	5.9	8.2 (Isoniazid)39.5 (Rifampicin)0.8 (Ethambutol)2.74 (Moxifloxacin)	2.9	−5.6 (Glide Score)	[48]
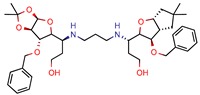 1	12	15.4 (Isoniazid ^1^)	46.2	−15.8 (Autodock)	[53]
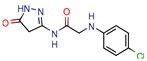 8b	0.4	0.03 (Pyrazinamide)0.08 (Ciprofloxacin)0.07 (Streptomycin)	-	NS (Autodock)	[54]
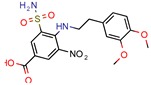 1	-	-	5.7	−20.3 (FlexX Score)	[60]
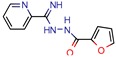 MB16695	67,8	37.2 (Isoniazid)	-	−6.6 kcal/mol (Glide)	[67]
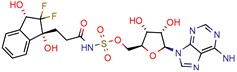 (1R,3S)-2	26.6	57.6 (AMS ^2^)	5	−11.9 (Glide Score)	[69]
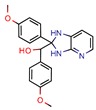 2j	38.5	0.5 (Ciprofloxacin)	-	−55.3 (Biopredicta Score)	[58]
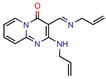 TB8	3.7	3.1 (Isoniazid)0.25 (Norfloxacin)	-	NS (FRIGATE)	[63]
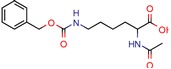 BTB13566	9.7	12.8 (Isoniazid ^1^)	-	−28.9 (FlexX Score)	[65]
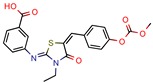 Lead 1a	56.75	72.8 (Isoniazid ^1^)	17.1	72.2 (GOLD Score)	[56]
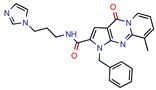 1	113.5	2.5 (Ampicilin)	-	NS (FRIGATE)	[55]
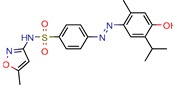 Conjugate-5	38.9	0.34 (Ampicilin)	-	−10.8 (Autodock VINA)	[59]
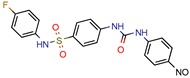 4e	44.14	122.6 (Isoniazid)	90	−6.8 kcal/mol (Glide)	[68]
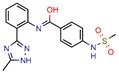 Compound 2	25.0	125.0 (Rifampicin)	-	−164.7 (CDOCKER)	[61]
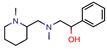 A	-	-	-	7.2 kcal/mol (Autodock)	[64] *
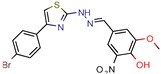 Compound 1	25.0	32.1 (Isoniazid)	6	Less than −40.0 (DOCK)	[57]
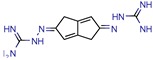 10	-	-	48	NS (SABRE)	[66]
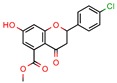 5c	81.9	5617.8 (Isoniazid)	25.34	−8.37 (Glide)	[62]
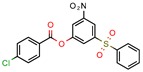 15	11.5	0.42 (Sulfometuron methyl)	1.85	Less than −7.0 (Glide Score)	[70]

* Manuscripts that performed other in vitro assays; ^1^ Ratio calculated using mean isoniazid as control; ^2^ AMS: 5′-O-sulfamoyl adenosine; NS: data not shown; NP: docking not performed.

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
