# Peer review of "Predictive Power of In Silico Approach to Evaluate Chemicals against M. tuberculosis: A Systematic Review"

_pharmaceuticals, 2019, doi:10.3390/ph12030135_

Round 1

Reviewer 1 Report

Timo et al, in this review, tried to gather the works and obtained results performed in the past, gathering the use of in silico approaches and in vitro validation in the development of new compounds active against Mycobacterium tuberculosis. The thematic addressed in this work is interesting and of utmost importance for further developments against tuberculosis, since the combination of in silico and in vitro approaches has shown to be vital to the development of innovative therapeutics for different diseases. Although the authors tried to focus on works performing screenings and consequent validation, they have missed many other works using, for example, other computational approaches that were shown to be very useful on identifying and characterizing newly identified compounds. As an example, let me call you attention of works performed my Dr Machuqueiro, Drª Martins and Dr Loewen (Vila-Vicosa, D., Victor, B.L., Ramos, J., Machado, D., Viveiros, M., Switala, J., Loewen, P.C., Leitão, R., Martins, F., Machuqueiro, M., (2017) "Insights on the mechanism of action of INH-C10 as an antitubercular prodrug", Mol. Pharm., 14, 4597-4605. ; Machuqueiro, M., Victor, B. L., Switala, J., Villanueva, J., Rovira, C., Fita, I., Loewen, P. C., (2017) "Catalase activity of catalase-peroxidases is modulated by changes in pKa of the distal histidine", Biochemistry , 56, 2271-2281. ; Teixeira, V.H., Ventura, C., Leitão, R., Rafols, C., Bosch, E., Martins, F., Machuqueiro, M., (2015) "Molecular details of INH-C10binding to wt KatG and to its S315T mutant", Mol. Pharmaceutics , 12, 898-909.), wherein the recent past years they have combined in silico and in vitro validation assays to identify and characterize new Mtb inhibitory compounds. And I believe that this is just one example of works not mentioned in the literature analyzed by the authors.

Although the paper is clear and well written, I believe that if the authors want to perform a real review regarding results obtained from the combination of in silico and in vitro approaches to identify new active compounds against tuberculosis, they should put more effort on the identification of all the bibliography regarding this area of research. Otherwise, I cannot see this work as a review as it is meant to be.

Reviewer 2 Report

The article entitled “Predictive power of in silico approach to evaluate chemicals against M. tuberculosis: a systematic review” submitted by Timo & Co-workers describes a systematic analysis of manuscripts using in silico methods to find antimycobacterial molecules and subsequent in vitro or in vivo testing. 

The review presents an overview of the in silico studies done against M. tuberculosis, in terms of enzymatic targets, virtual screening methods, virtual screening databases, docking software, in vitro/in vivo testing, etc. As researcher with interest in M. tuberculosis and experience in these fields, I find the purpose of this review interesting and very useful. However, the study has a number of aspects that need to be clarified and corrected, before being ready for publication.

MAJOR PROBLEMS

(1)   The representability of the articles selected. The studied identified 2424 articles. Through a variety of screening and eligibility processes, only 45 were included in the analysis. The choice of these 45 articles is a critical aspect that affects the outcome of the analysis, especially with such significant cuts. So, it has to be made very clearly. 

1.1 It is strange that the articles retrieved from PubMed (632), Web of Science (929) and Science Direct (863), when combined and after duplicate removal still give 1973 articles. This would mean that the number of articles unique among these three databases would be very high, while in reality the majority of them should be repeated. This should be checked.

1.2 It is not clear how the screening procedure of the database was performed (from 1973 to 237 articles). The authors need to explicitly define the eligibility and exclusion criteria. 

1.3 It is not clear the passage from 237 to 72 articles, nor from these to 40 (+5 addition). This should be better explained. 

(2)   The article has a number of tables that could be complemented or in some cases replaced by figures with pie charts. E.g. Table 1, 2, 3, and 4.

(3)   Table 1. Within the enzyme targets identified, the results could be sorted by the specific resolution of each entry.

(4)   Databases Screened. The authors describe the ZINC database as having 120 million compounds. However, normally no VS study uses this number. Large VS runs apply the clean LEAD subset database of ZINC, with 4.6 million compounds, or other smaller databases. The authors should clarify this issue.

(5)   Docking software. VINA is one of the most widely used docking programs and approximately 100 times faster than Autodock. Both programs were developed by the same research group (but by different people) and are widely used. Their official name is AutoDock 4 and AutoDock Vina, but among the community there are often called Autodock and VINA, respectively. However, they use completely different approximations and scoring functions and co-exist until today, each with their strengths and weaknesses.  The authors should clarify in Table 4, which program is actually mentioned in the papers (Autodock or VINA, or a mixture of the both).

(6)   Table 5. The compound name expressed as 2, 4h, 5b, etc has no meaning in the review. It forces readers to go back to the original paper to understand the present table. Authors could include a chemdraw representation of the molecules indicated.

(7)   Table 5 – docking score columm. There are several problems with the information presented. The autodock (or vina) score are always negative. So a value of 14.4 as presented in line 2 would be impossible. A value of -14.4 kcal/mol would also be a bit too negative for both vina or autodock 4. Several entries are described as GOLD, but the gold software program has 4 different scoring functions (Chemscore, goldscore, chemplp, asp) each working on different scoring ranges. The exact scoring function of GOLD should be indicated.

(8)   Figure 1 – the chart can be greatly improved. In particular, it would be interesting to use a double entry line accompanying the evolution of the number of virtual screening articles in general. 

(9)   The authors could improve the level of the review by presenting a structural analysis of the binding pocket of the 29 enzyme targets described, as well as a description of their function. 

(10)       The authors should improve their introduction on docking and virtual screening to make the work more accessible for non specialists in in-silico work.

Round 2

Reviewer 1 Report

I am satisfied with the answers given to my comments. Therefore, I believe that I can suggest the work for publication.

Reviewer 2 Report

The authors have taken into account all the issues raised and have made all the required changes.